# Comparison of RIPASA and ALVARADO scores for risk assessment of acute appendicitis: A systematic review and meta-analysis

Giuliana Favara[1], Andrea Maugeri[1], Martina Barchitta[1], Andrea Ventura[2], Guido Basile[2], Antonella Agodi [1]*

1 Department of Medical and Surgical Sciences and Advanced Technologies "GF Ingrassia", University of Catania, Catania, Italy, 2 Department of General Surgery and Medical-Surgical Specialties, University of Catania, Catania, Italy

* agodia@unict.it

**Data Availability Statement:** All relevant data are within the paper and its Supporting Information files.

## Abstract

### Background

In the last decades, several clinical scores have been developed and currently used to improve the diagnosis and risk management of patients with suspected acute appendicitis (AA). However, some of them exhibited different values of sensitivity and specificity. We conducted a systematic review and metanalysis of epidemiological studies, which compared RIPASA and Alvarado scores for the diagnosis of AA.

### Methods

This systematic review was conducted using PubMed and Web of Science databases. Selected studies had to compare RIPASA and Alvarado scores on patients with suspected AA and reported diagnostic parameters. Summary estimates of sensitivity and specificity were calculated by the Hierarchical Summary Receiver Operating Curve (HSROC) using STATA 17 (STATA Corp, College Station, TX) and MetaDiSc (version 1.4) software.

### Results

We included a total of 33 articles, reporting data from 35 studies. For the Alvarado score, the Hierarchical Summary Receiver Operating Curve (HSROC) model produced a summary sensitivity of 0.72 (95%CI = 0.66–0.77), and a summary specificity of 0.77 (95%CI = 0.70–0.82). For the RIPASA score, the HSROC model produced a summary sensitivity of 0.95 (95%CI = 0.92–0.97), and a summary specificity of 0.71 (95%CI = 0.60–0.80).

### Conclusion

RIPASA score has higher sensitivity, but low specificity compared to Alvarado score. Since these scoring systems showed different sensitivity and specificity parameters, it is still necessary to develop novel scores for the risk assessment of patients with suspected AA.

**Funding:** The authors received no specific funding for this work.

**Competing interests:** The authors have declared that no competing interests exist.

## Introduction

Acute appendicitis (AA) represents one of the most frequent disorders in abdominal surgery, with a prevalence ranging from 7 to 12% in the general population [1, 2]. If untreated or undiagnosed, AA could lead to a higher risk of adverse outcomes, including death. Despite its common occurrence, the diagnosis of AA is still challenging for clinicians, suggesting the need of novel approaches to improve patients' management [3, 4]. Indeed, clinical presentation of AA is commonly atypical and easily mistaken for other conditions, with only about 40% of the cases presenting typical signs and symptoms (i.e., periumbilical pain, nausea, vomiting, pain migration to the right lower quadrant) [5–7].

In the last decades, several scoring systems have been developed to assist clinicians in the assessment of patients with suspected appendicitis [8, 9]. Among these, the ALVARADO score —proposed for the first time in 1986—is one of the most widely used in the diagnosis of AA based on 6 clinical parameters and 2 laboratory measurements (i.e., localized tenderness in the right lower quadrant, migration of pain, temperature elevation, nausea-vomiting, anorexia, rebound pain, leukocytosis and leukocyte shift to the left) [8]. Despite not being specific enough, a score of 4–5 is compatible with the diagnosis of AA, a score of 7–8 indicates a probable appendicitis, and a score of 9–10 indicates a very probable AA [10, 11]. However, the Alvarado score is also considered lacking some parameters, including age, gender, and duration of symptoms, which have shown to be crucial in the diagnosis of AA [3, 12]. The RIPASA is one of the most recently developed scoring systems, which is based on six additional clinical and personal patients' parameters than those included in the Alvarado score (i.e., age, gender, duration of symptoms, guarding, Rovsing's sign, and negative urinalysis).

In this case, a RIPASA score of more than 7.5 is considered positive for appendicitis [1, 8, 11, 13–15]. Although RIPASA and Alvarado scores are the most commonly used in clinical practice, no clear indication exists for choosing what scoring system might be more suitable for patients at risk of AA [16]. Here, we conducted a systematic review and metanalysis of epidemiological studies comparing RIPASA and Alvarado scores, in order to identify which is the one providing more accurate diagnosis of AA.

## Material and methods

### Literature search and selection criteria

The current systematic review was conducted in accordance with the Preferred Reporting Items for Systematic Reviews and Meta-analyses (PRISMA) statements and the Cochrane Handbook's guidelines (PRISMA checklist available in **S1 Appendix**) [17]. The research protocol was registered in the PROSPERO database, with the code CRD42022339490. Two authors (GB and AV) conducted a literature search of articles, using the databases PubMed and Web of Science. The electronic search strategy included the following keywords: ((Appendicitis) AND (RIPASA) AND (Alvarado)). The last search was conducted on 21 July 2021. After identifying and removing duplicates, the authors also conducted a cross-search through the articles cited by the studies, aiming to identify additional articles to be included in the systematic review. Selected studies had to meet the following inclusion criteria: (i) observational studies; (ii) which provided full-text and written in English language; (iii) which included patients with suspected acute appendicitis (iv) and compared RIPASA and Alvarado scores. By contrast, the following articles were excluded: (i) experimental studies; (ii) studies conducted only on a specific population (e.g. pregnant women or pediatric patients); (iii) studies not comparing the mentioned scoring systems; (iv) studies conducted on patients with an already

established cause of abdominal pain and/or patients who experienced pain for a prolonged period; (v) letters, comments, case reports, case series, reviews.

Titles and abstracts of all identified articles were independently screened by two authors (GB and AV). Articles potentially eligible were full-text reviewed to assess whether eligibility criteria were fully met. Discordant opinions between investigators were resolved by consulting a third author (AA).

## Data extraction

The following information was extracted from all included studies: first author, year of publication, study design, sample size, age, sex, histologically confirmed acute appendicitis, other previous diagnoses, computerized tomography (CT) performed. In addition, for both the RIPASA and Alvarado scores, the authors collected the following information: specificity, sensitivity, positive predictive value, negative predictive value, diagnostic accuracy, negative appendicectomy rate, area under the roc-curve, positive likelihood ratio, negative likelihood ratio. Discordant opinions between investigators were resolved by consulting a third author (AA).

## Definitions of RIPASA and ALVARADO scores

Clinical Scoring Systems are useful to group patients according to their symptoms and signs, and to identify patients with suspected appendicitis. Alvarado clinical score includes 6 clinical parameters and 2 laboratory measurements, which are relevant in the diagnosis of acute appendicitis. Among these, migration of abdominal pain to the right iliac fossa, anorexia or ketones in the urine, nausea or vomiting, localized tenderness in the right iliac fossa, rebound pain, body temperature more than 37.3˚C, leukocytosis, and neutrophilia. Alvarado score indicates a confirmed, probable, or very probable diagnosis of acute appendicitis, in the case of a score of 4–6, 7–8, or 9–10, respectively. Commonly, a score of 7.0 is considered as positive for appendicitis [10, 11].

RIPASA clinical score includes the following parameters: age, gender, right iliac fossa pain, migration of pain to the right iliac fossa, nausea or vomiting, anorexia, duration of symptoms, localized tenderness in the right iliac fossa, guarding, rebound tenderness, Rovsing' s sign, fever, raised white cell count, negative urinalysis, and foreign national registration identity card. Commonly, a score above 7.5 is considered as positive for the diagnosis of appendicitis [1, 8, 11, 13–15].

## Risk of bias and quality assessment

The methodological quality of the included studies was assessed using a set of criteria for the Quality Assessment of Diagnostic Accuracy Studies (QUADAS-2). By considering 4 domains (i.e., patient selection, index test, reference standard, and flow and timing), this approach is useful for the evaluation of diagnostic accuracy studies. In particular, the questions can be answered using "low", "high" or "unclear" to judge the risk of bias [18].

## Statistical analysis

Meta-analysis of diagnostic test accuracy requires a statistically rigorous approach based on hierarchical models that respect the binomial data structure. In the present study, we first obtained for each score the forest plots of sensitivity and specificity and their 95% Confidence Intervals (CI) based on a random-effects model and using the MetaDiSc software (version 1.4). The heterogeneity was assessed with the $I^2$ statistic. Next, the summary estimates of

sensitivity and specificity were calculated by the Hierarchical Summary Receiver Operating Curve (HSROC), using the package Metandi for STATA 17 statistical software (STATA Corp, College Station, TX). To visualize the HSROC curve, we also used the command metandiplot.

## Results

### Selection and characteristics of included studies

**Fig 1** reported the PRISMA flow diagram describing the study selection process. A total of 75 studies were identified from the literature search, of which 53 were screened after removing duplicates. After full-text screening of 31 articles deemed eligible for inclusion, 2 studies not comparing two scoring systems considered, 2 reviews, and 1 study not written in English were excluded. After a cross-search through the articles cited by the studies, the authors identified 7 additional articles to be included. Hence, a total of 33 studies were included in the present systematic review and meta-analysis. However, Abdelrhman et al. (2018) reported findings from two different populations, while Erdem et al. (2013) used two different couples of cut-offs for the RIPASA and ALVARADO scores. Accordingly, the meta-analysis was conducted on 35 different estimates of sensitivity and specificity. **Table 1** shows the main characteristics (i.e., country, type of study, sample size) of the included studies, as well as characteristics of patients (i.e., age, sex). **Table 2**, instead, summarizes statistical parameters of RIPASA and ALVA-RADO scores, respectively.

### Main characteristics of included studies

All the included studies were published between 2011 and 2020. In particular, most of the studies were conducted in South-Eastern countries, of which 14 in India, 5 in Turkey, 2 in Pakistan, 2 in Egypt, 2 in Iran, 1 in Jordan, 1 in China, 1 in Korea, 1 in Brunei, 2 in Mexico, 1 in USA and 1 in Poland. With respect to the study design, all the 33 articles included in the study were observational studies. Specifically, 26 were prospective, 4 retrospective, and 3 cross-sectional. The overall sample size ranged from 56 to 600 participants. Although gender distribution throughout the studies was fairly balanced, almost all studies reported a higher proportion of men. The most commonly considered symptom to identify patients with AA was the pain in Right Iliac Fossa. Moreover, some studies required more extensive list of clinical symptoms, as well as advanced imaging techniques.

### Cut-offs of scoring systems

In the various studies, diagnostic parameters for RIPASA and Alvarado scores were calculated according to different cut-offs. Most of the studies used 7.0 and 7.5 as conventional cut-offs for Alvarado and RIPASA scores, respectively. Accordingly, patients were considered as affected by AA if their scores exceeded these cut-off values. However, Korkut et al. and Ozdemir et al. used the value of 8 for the Alvarado, and the values of 10 or 12 for the RIPASA, respectively. Reasons of using different cut-offs may be explained by the aim to improve the diagnostic parameters of the scores. For all the studies considered, the gold standard is given by the histopathological exam performed post-surgery.

### Scoring systems performances

Overall, the present systematic review included 5384 patients with AA who were tested with the RIPASA and Alvarado scores. The sensitivity values ranged from 16.4% to 100% for the RIPASA score, and from 14.8% to 97.2% for the Alvarado score (**Fig 2**). Interestingly, all

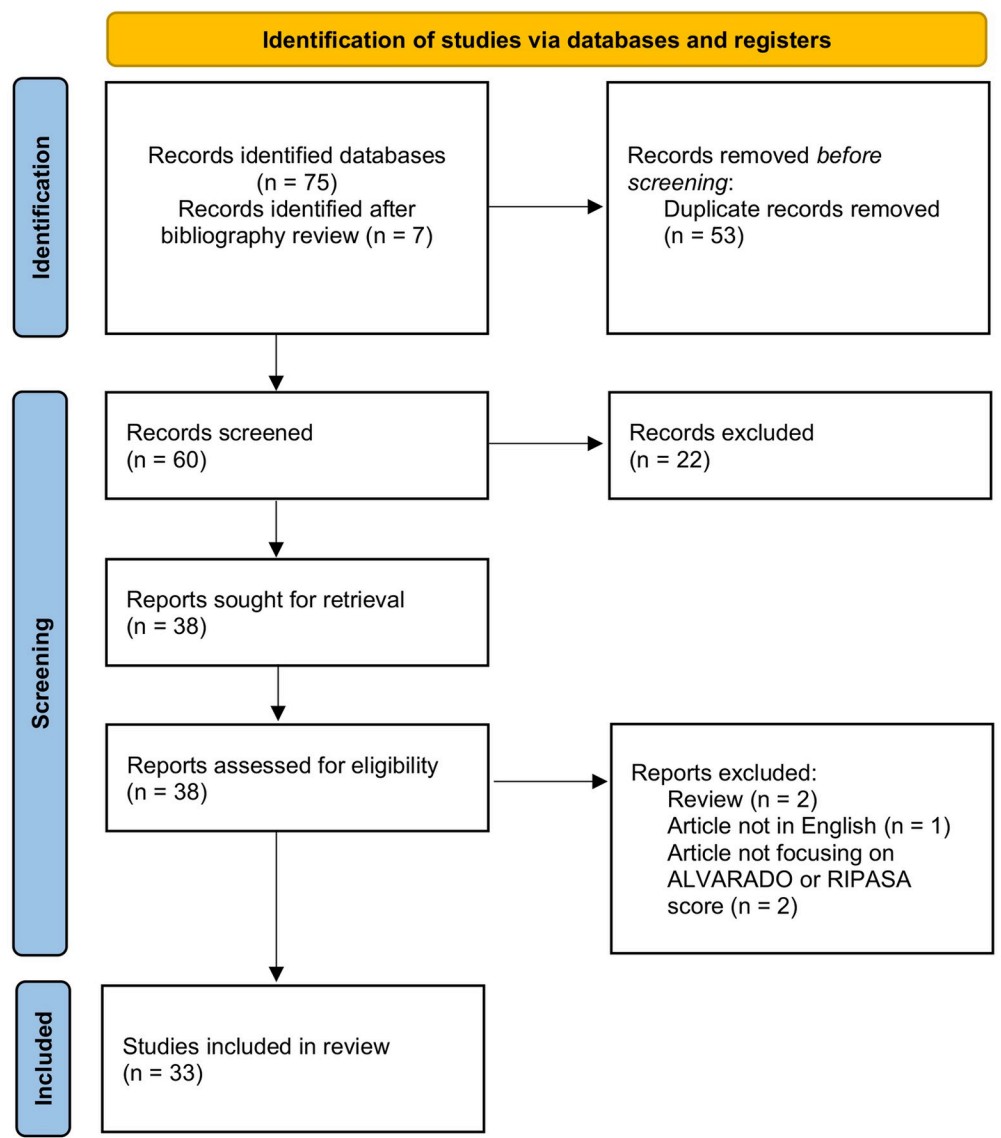

From: Page MJ, McKenzie JE, Bossuyt PM, Boutron I, Hoffmann TC, Mulrow CD, et al. The PRISMA 2020 statement: an updated guideline for reporting systematic reviews. BMJ 2021;372:n71. doi: 10.1136/bmj.n71

For more information, visit: http://www.prisma-statement.org/

**Fig 1. Prisma flow diagram describing study selection process.**

studies reported a higher sensitivity for the RIPASA score than for the Alvarado score. Most of the studies reported higher values of specificity for the Alvarado score than for the RIPASA score. The specificity values ranged from 9% to 100% for the RIPASA score, and from 16% to 100% for the Alvarado score (**Fig 3**). The majority of studies reported higher Positive Predictive Value for the Alvarado score. Conversely, the majority of studies reported higher Negative Predictive Values for the RIPASA score. Moreover, in the studies included in the present

**Table 1. Characteristics of studies included in the systematic review.**

| Study | Country | Study design | Sample size | Age (years) | Sex (% of male) | Histologically diagnosis of AA |
|---|---|---|---|---|---|---|
| Noor et al. 2020 [1] | Pakistan | Prospective | 300 | Mean = 28; SD = 10.0 | 58.7% | 270 |
| Dezfuli et al. 2020 [10] | Iran | Prospective | 133 | Mean = 28.3; SD = 4.8 | 55.6% | 76 |
| Korkut et al. 2020 [22] | Turkey | Prospective | 74 | Mean = 36.68; SD = 11.97 | 56. 8% | 65 |
| Şenocak et al. 2020 [9] | Turkey | Retrospective | 202 | Mean = 25.6; SD = 8.8 | 75.2% | 170 |
| Devarajan et al. 2019 [23] | India | Prospective | 250 | NA | 66.0% | 237 |
| Ozdemir et al. 2019 [24] | Turkey | Retrospective | 76 | Mean = 33.8; SD = 13.2 | 57.9% | 59 |
| Ak et al. 2019 [25] | Turkey | Prospective | 218 | Median = 33 | 48.2% | 107 |
| Akbar et al. 2019 [11] | Pakistan | Prospective | 288 | NA | 57.3% | 252 |
| Bolivar-Rodriguez et al. 2018 [26] | Mexico | Prospective | 137 | NA | - | 108 |
| Ansara et al. 2018 [27] | India | Prospective | 100 | Mean = 32.16 | 57.0% | 74 |
| Patil et al. 2018 [28] | India | Prospective | 150 | NA | 69.3% | NA |
| Chavan et al. 2018 [29] | India | Cross-sectional | 100 | NA | 71.0% | 99 |
| Abdelrhman et al. 2018a [30] | Egypt | Prospective | 100 | NA | 40.0% | 89 |
| Abdelrhman et al. 2018b [30] | Egypt | Prospective | 100 | NA | 59.0% | 82 |
| Pasumarthi et al. 2018 [31] | India | Prospective | 116 | Mean = 34.4 | 48.3% | 96 |
| Elhosseiny et al. 2018 [32] | Egypt | Cross-sectional | 56 | Mean = 28.3; SD = 8.1 | 35.7% | 46 |
| Nancharaiah et al. 2018 [33] | India | Prospective | 150 | NA | - | 144 |
| Arroyo-Rangel et al. 2017 [34] | Mexico | Prospective | 100 | Mean = 36.5; SD = 16.2 | 42.0% | 85 |
| Rodrigues et al. 2017 [35] | India | Prospective | 105 | NA | 45.7% | 86 |
| Karami et al. 2017 [36] | Iran | Prospective | 100 | Mean = 32; SD = 10 | 66.0% | 88 |
| Chae et al. 2017 [37] | Korea | Retrospective | 189 | NA | 33.3% | 61 |
| Regar et al. 2017 [38] | India | Prospective | 100 | Mean = 24.86 | 61.0% | 95 |
| Subramani et al. 2017 [39] | India | Prospective | 96 | Mean = 30.58; SD = 12.3 | 47.9% | 50 |
| Golden et al. 2016 [5] | USA | Prospective | 287 | Median = 31; IQR = 12–88 | 40.1% | NA |
| Muduli et al. 2016 [40] | India | Prospective | 96 | Mean = 23.5; SD = 9.42 | 72.9% | 73 |
| Sinnet et al. 2016 [41] | India | Cross-sectional | 109 | Mean = 28 | 36.7% | 89 |
| Liu et al. 2015 [42] | China | Retrospective | 297 | Mean = 47.9; SD = 17.6 | 53.2% | 187 |
| Srikantaiah et al. 2015 [43] | India | Prospective | 150 | Mean = 25.87 | 69.3% | 111 |
| Verma et al. 2015 [44] | India | Prospective | 100 | Mean = 28.10±10.88 | 67.0% | 91 |
| Walczak et al. 2015 [45] | Poland | Prospective | 94 | Mean = 38 | 51.1% | 59 |
| NaNjuNdaiah et al. 2014 [46] | India | Prospective | 206 | Mean = 27.82; SD = 9.26 | 61.7% | 184 |
| Erdem et al. 2013 [47] | Turkey | Prospective | 113 | Mean = 30.2; SD = 10.1 | 54.9% | 77 |
| Alnjadat et al. 2013 [48] | Jordan | Prospective | 600 | Mean = 26.52 | 60.0% | 498 |
| Chong et al.2011 [49] | Brunei | Prospective | 192 | Mean = 25.1; SD = 12.7 | 47.9% | 101 |

meta-analysis, the RIPASA score showed higher values of diagnostic accuracy and Area Under the Curve (AUC) compared to the Alvarado.

**Fig 4** shows hierarchical summary estimates of sensitivity and specificity for the Alvarado and the RIPASA scores, respectively. The graphs also report a 95% prediction ellipse for the individual values of sensitivity and specificity, and the 95% confidence ellipse around the mean values of sensitivity and specificity. For the Alvarado score (**Fig 4A**), the HSROC model produced a summary sensitivity of 0.72 (95%CI = 0.66–0.77), and a summary specificity of 0.77 (95%CI = 0.70–0.82). The heterogeneity was $I^2 = 0.90$ for the sensitivity and $I^2 = 0.59$ for the specificity. For the RIPASA score (**Fig 4B**), the HSROC model produced a summary sensitivity of 0.95 (95%CI = 0.92–0.97), and a summary specificity of 0.71 (95%CI = 0.60–0.80). The heterogeneity was $I^2 = 0.76$ for the sensitivity and $I^2 = 0.70$ for the specificity.

**Table 2. Characteristics of clinical scoring systems for each study included in the systematic review.**

| Study | Score | Cut–off | Sensitivity (%) | Specificity (%) | PPV (%) | NPV % | Diagnostic Accuracy (%) | Negative appendectomy rate (%) | AUC |
|---|---|---|---|---|---|---|---|---|---|
| Noor et al. 2020 | Ripasa | 7.5 | 98.5 | 90 | 98.9 | 87.1 | 97.7 | 10 | NA |
| Noor et al. 2020 | Alvarado | 7 | 68.1 | 80 | 96.8 | 21.8 | 69.3 | 20 | NA |
| Dezfuli et al. 2020 | Ripasa | 7.7 | 93.4 | 45.6 | 69.6 | 83.9 | NA | NA | 0.739 |
| Dezfuli et al. 2020 | Alvarado | 6 | 53.9 | 70.2 | 70.7 | 53.3 | NA | NA | 0.662 |
| Korkut et al. 2020 | Ripasa | 12 | 75 | 99.7 | 98.0 | 34.8 | NA | NA | 0.893 |
| Korkut et al. 2020 | Alvarado | 8 | 60.9 | 89.9 | 97.6 | 24.2 | NA | NA | 0.938 |
| Şenocak et al. 2020 | Ripasa | 9.8 | 83.5 | 37.5 | 87.6 | 30 | NA | 12.3 | 0.605 |
| Şenocak et al. 2020 | Alvarado | 7.3 | 75.8 | 65.6 | 92.1 | 33.8 | NA | 7.9 | 0.708 |
| Devarajan et al. 2019 | Ripasa | 7.5 | 98.4 | 90 | 99.5 | 75 | 97 | NA | NA |
| Devarajan et al. 2019 | Alvarado | 7 | 73.7 | 80 | 94.3 | 3.4 | 74 | NA | NA |
| Ozdemir et al. 2019 | Ripasa | 10 | 68 | 71 | 89 | 39 | 75 | NA | 0.700 |
| Ozdemir et al. 2019 | Alvarado | 8 | 36 | 82 | 87 | 27 | 56 | NA | 0.600 |
| Ak et al. 2019 | Ripasa | 7.5 | 91.6 | 65.8 | NA | NA | 0.9 | 14.3 | 0.880 |
| Ak et al. 2019 | Alvarado | 5 | 72.9 | 54.1 | NA | NA | 0.7 | 71.4 | 0.710 |
| Akbar et al. 2019 | Ripasa | 7.5 | 98 | 75 | 96.5 | 84.7 | NA | NA | NA |
| Akbar et al. 2019 | Alvarado | 7 | 53 | 75 | NA | NA | NA | NA | NA |
| Bolìvar-Rodriguez et al. 2018 | Ripasa | 7.5 | 97.2 | 27.6 | 83.3 | 72.7 | 82.5 | NA | NA |
| Bolìvar-Rodriguez et al. 2018 | Alvarado | 7 | 97.2 | 27.6 | 83.3 | 72.7 | 82.5 | NA | NA |
| Ansara et al. 2018 | Ripasa | 7.5 | 91.9 | 80.8 | 93.2 | 77.8 | 89 | 6.8 | NA |
| Ansara et al. 2018 | Alvarado | 7 | 68.9 | 73.1 | 87.9 | 45.2 | 70 | 12.1 | NA |
| Patil et al. 2018 | Ripasa | 7.5 | 95.5 | 89.7 | 95 | 89 | NA | NA | 0.926 |
| Patil et al. 2018 | Alvarado | 7 | 81.1 | 87.2 | 81 | 87 | NA | NA | 0.841 |
| Chavan et al. 2018 | Ripasa | 7.5 | 90.8 | 100 | 100 | 10 | 90 | 0 | NA |
| Chavan et al. 2018 | Alvarado | 7 | 75.8 | 100 | 100 | 4 | 76 | 0 | NA |
| Abdelrhman et al. 2018a | Ripasa | 7.5 | 95.5 | 72.7 | 96.6 | 66.7 | 93 | NA | 0.950 |
| Abdelrhman et al. 2018a | Alvarado | 7 | 73 | 81.8 | 97 | 27.3 | 74 | NA | 0.740 |
| Abdelrhman et al. 2018b | Ripasa | 7.5 | 97.6 | 66.7 | 93 | 85.7 | 92 | NA | 0.870 |
| Abdelrhman et al. 2018b | Alvarado | 7 | 79.3 | 83.3 | 95.6 | 46.9 | 80 | NA | 0.860 |
| Pasumarthi et al. 2018 | Ripasa | 7.5 | 75 | 65 | 91.1 | 35.1 | 73.3 | NA | 0.810 |
| Pasumarthi et al. 2018 | Alvarado | 6 | 52.1 | 80 | 92.6 | 25.8 | 56.9 | NA | 0.771 |
| Elhosseiny et al. 2018 | Ripasa | 7.5 | 100 | 75 | 95.8 | 100 | 88 | 4.2 | NA |
| Elhosseiny et al. 2018 | Alvarado | 7 | 65.2 | 100 | 100 | 33.3 | 83 | 0 | NA |
| Nancharaiah et al. 2018 | Ripasa | 7.5 | 98.6 | 83.3 | 93.3 | 71.4 | NA | NA | 0.892 |
| Nancharaiah et al. 2018 | Alvarado | 7 | 76.4 | 66.7 | 89 | 10.5 | NA | NA | 0.757 |
| Arroyo-Rangel et al. 2017 | Ripasa | NA | 99 | 71 | 96 | 91 | NA | NA | 0.880 |
| Arroyo-Rangel et al. 2017 | Alvarado | NA | 91 | 64 | 94 | 60 | NA | NA | 0.800 |
| Rodrigues et al. 2017 | Ripasa | 7.5 | 93 | 31.6 | 86 | 50 | NA | NA | NA |
| Rodrigues et al. 2017 | Alvarado | 7 | 81.4 | 47.4 | 87.5 | 36 | NA | NA | NA |
| Karami et al. 2017 | Ripasa | 8 | 93.2 | 91.7 | 98.8 | 64.7 | NA | NA | 0.980 |
| Karami et al. 2017 | Alvarado | 7 | 78.4 | 100 | 100 | 38.7 | NA | NA | 0.910 |
| Chae et al. 2017 | Ripasa | 7.5 | 16.4 | 99.2 | 90.9 | 71.3 | 65.3 | NA | 0.650 |
| Chae et al. 2017 | Alvarado | 7 | 14.8 | 95.3 | 60 | 70.1 | 69.8 | NA | 0.700 |
| Regar et al. 2017 | Ripasa | 7.5 | 94.7 | 60 | 97.8 | 37.5 | 93 | 2.2 | NA |
| Regar et al. 2017 | Alvarado | 7 | 67.4 | 80 | 98.5 | 11.4 | 68 | 1.5 | NA |
| Subramani et al. | Ripasa | 7.5 | 98 | 80.4 | 84.4 | 97.4 | 89.6 | 15.5 | NA |

*(Continued)*

**Table 2.** (Continued)

| Study | Score | Cut–off | Sensitivity (%) | Specificity (%) | PPV (%) | NPV % | Diagnostic Accuracy (%) | Negative appendectomy rate (%) | AUC |
|---|---|---|---|---|---|---|---|---|---|
| Subramani et al. | Alvarado | 7 | 68 | 86.9 | 85 | 71.4 | 77.1 | 15 | NA |
| Golden et al. 2016 | Ripasa | 7.5 | 78 | 36 | 39 | 76 | NA | NA | 0.670 |
| Golden et al. 2016 | Alvarado | 7 | 61 | 74 | 53 | 79 | NA | NA | 0.720 |
| Muduli et al. 2016 | Ripasa | 7.5 | 97.3 | 75 | 89.9 | 92.3 | 90.5 | NA | NA |
| Muduli et al. 2016 | Alvarado | 7 | 68.5 | 84.4 | 90.1 | 54 | 73.3 | NA | NA |
| Sinnet et al. 2016 | Ripasa | 7.5 | 95.5 | 65 | 92.4 | 76.5 | 89.9 | 7.6 | 0.943 |
| Sinnet et al. 2016 | Alvarado | 7 | 65.2 | 90 | 96.7 | 36.7 | 69.7 | 3.3 | 0.862 |
| Liu et al. 2015 | Ripasa | 7.5 | 95.2 | 73.6 | NA | NA | 87.2 | NA | NA |
| Liu et al. 2015 | Alvarado | 7 | 63.1 | 80.9 | NA | NA | 69.7 | NA | NA |
| Srikantaiah et al. 2015 | Ripasa | 7.5 | 95.5 | 89.7 | 95 | 89 | NA | NA | 0.926 |
| Srikantaiah et al. 2015 | Alvarado | 7 | 81.1 | 87.2 | 81 | 87 | NA | NA | 0.841 |
| Verma et al. 2015 | Ripasa | 7.5 | 100 | 11.1 | 91.9 | 100 | 92 | 8.1 | NA |
| Verma et al. 2015 | Alvarado | 7 | 82.4 | 44.4 | 93.7 | 20 | 79 | 6.3 | NA |
| Walczak et al. 2015 | Ripasa | 7.5 | 88 | 9 | 68 | 20 | NA | NA | NA |
| Walczak et al. 2015 | Alvarado | 7 | 85 | 16 | 74 | 29 | NA | NA | NA |
| NaNjuNdaiah et al. 2014 | Ripasa | 7.5 | 96.2 | 90.5 | 98.9 | 73.1 | 96.2 | NA | 0.982 |
| NaNjuNdaiah et al. 2014 | Alvarado | 7 | 58.9 | 85.7 | 97.3 | 19.1 | 58.9 | NA | 0.849 |
| Erdem et al. 2013a | Ripasa | 7.5 | 100 | 28 | 75 | 100 | 77 | 25 | 0.857 |
| Erdem et al. 2013a | Alvarado | 7 | 82 | 75 | 88 | 66 | 80 | 12 | 0.818 |
| Erdem et al. 2013b | Ripasa | 10.2 | 83 | 75 | NA | NA | NA | NA | 0.857 |
| Erdem et al. 2013b | Alvarado | 6.5 | 82 | 75 | NA | NA | NA | NA | 0.818 |
| Alnjadat et al. 2013 | Ripasa | 7.5 | 93.2 | 61.8 | 92.2 | 64.9 | 91.5 | 7.8 | 0.914 |
| Alnjadat et al. 2013 | Alvarado | 7 | 73.7 | 68.6 | 92 | 34.8 | 74.3 | 8 | 0.743 |
| Chong et al.2011 | Ripasa | 7.5 | 98 | 81.3 | 85.3 | 97.4 | 91.8 | 14.7 | 0.918 |
| Chong et al.2011 | Alvarado | 7 | 68.3 | 87.9 | 86.2 | 71.4 | 86.5 | 13.8 | 0.865 |

## Quality assessment

The details of the quality assessment are reported in **S2 Appendix**. In general, the risk of bias was unclear or high for all domains under investigation (i.e., patient selection, index test, reference standard, and flow and timing). Similarly, we noted unclear or high concerns of applicability for all studies.

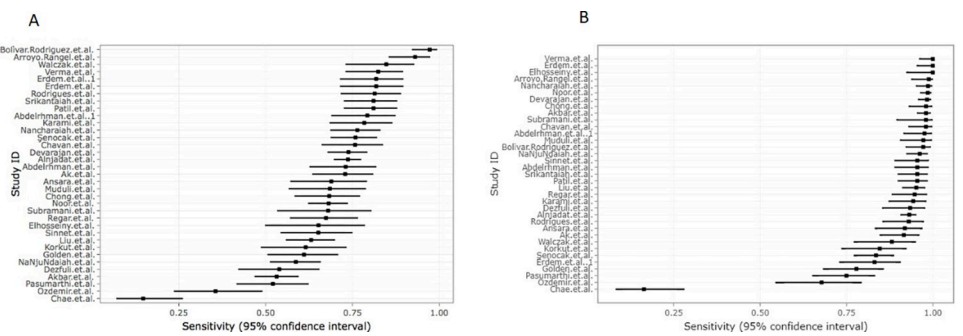

**Fig 2.** Plots of individual values of sensitivity for the Alvarado (A) and RIPASA (B) scores.

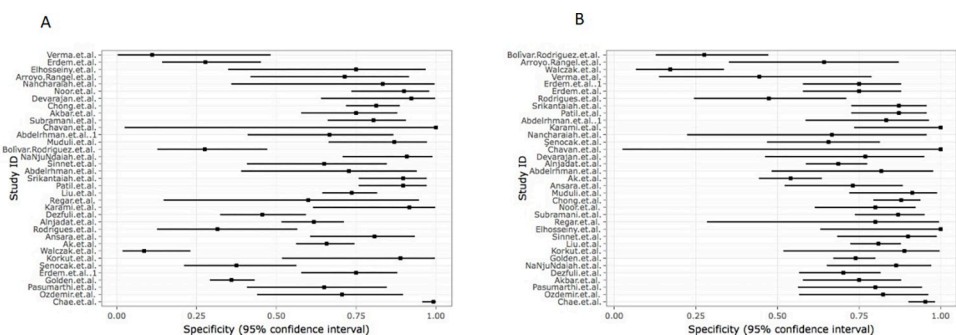

**Fig 3.** Plots of individual values of specificity for the Alvarado (A) and RIPASA (B) scores.

## Discussion

AA is one of the most common causes of acute abdominal pain, posing a serious diagnostic challenge for general surgeons due to its clinical variability and high prevalence [3]. Although a wide range of diagnostic tests hold great promise in clinical practice, early identifying an abnormal appendicitis is still challenging both for avoiding unnecessary surgical intervention and reducing healthcare costs [19, 20]. Moreover, complications related to the inflammation of the appendix further complicate patient's prognosis, also suggesting the need of implementing prediction scoring systems [20]. In this scenario, the use of clinical scoring systems can help healthcare providers in improving decision-making, patients' management, and identification of suspected appendicitis [3]. Moreover, several lines of evidence suggest that the integrated use of clinical scoring systems and diagnostics images allow to correctly identify patients with AA [3, 8]. Among the most common scores, RIPASA and Alvarado constitute the most utilized to clinically diagnose appendicitis in suspected patients [21]. In this study, we carried out a systematic review and meta-analysis of epidemiological studies comparing these two scores in terms of sensitivity and specificity. In line with previous evidence, our results reveal that the RIPASA score has higher sensitivity but lower specificity than the Alvarado

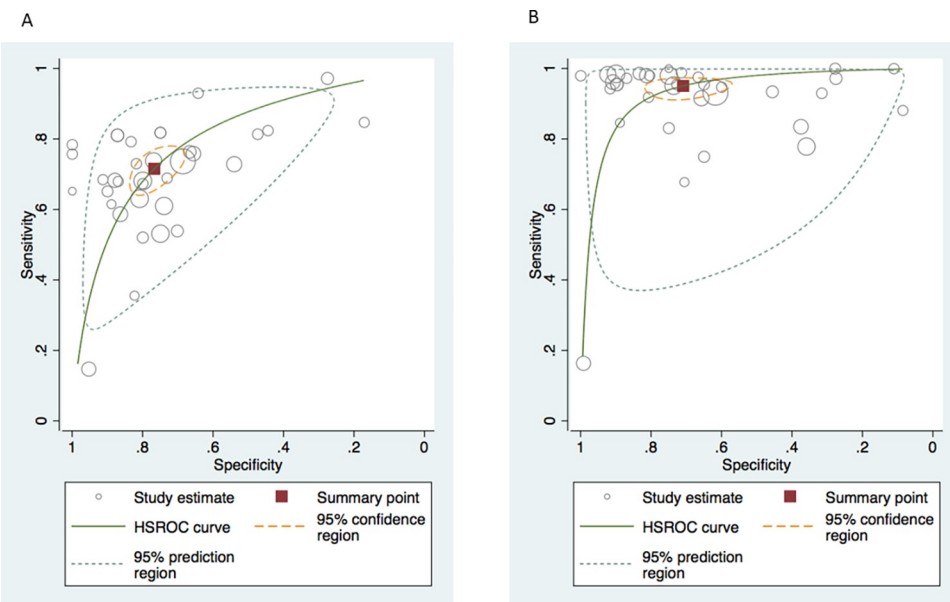

**Fig 4.** The HSROC of sensitivity and specificity for the Alvarado (A) and RIPASA (B) scores.

score. It means that the RIPASA score has a higher ability in predicting patients with AA, but also giving a high proportion of false positives. Thus, these findings should be considered when choosing the most appropriate test for the clinical practice. On the one hand, the high diagnostic performance of the RIPASA score could reduce the morbidity and mortality of patients with AA. On the other hand, however, the high number of false positives could lead to an increase in inappropriate procedures and healthcare costs.

To our knowledge, the strength of our work was represented by the lack of systematic reviews and meta-analyses in medical literature published on the same topic. Moreover, our study considered two scoring systems that have the advantages of being easy to use for clinicians, also requiring low healthcare costs to be applied. However, our study had some limitations to be considered. Firstly, most studies included in the present meta-analysis considered different cut-off values for the RIPASA and Alvarado scoring systems. Therefore, this could be considered a potential source of bias, also increasing the heterogeneity between studies. In fact, our analysis detected significant heterogeneity for both sensitivity and specificity. The quality assessment also reported an unclear-high risk of bias associated to patient selection, index test, reference standard, flow, and timing. Another source of misinterpretation is the possible existence of publication bias, which occurs when some studies have a higher probability to be published than others. However, there are no currently adequate methods to detect publication bias in meta-analyses of diagnostic tests, not allowing to completely exclude the presence of this kind of bias. Secondly, these scoring systems are mainly based on patient' clinical parameters measured in emergency situations and critical environments, which in turn could lead to wrong diagnoses and scoring systems calculation. Moreover, using these two scores could make difficult the diagnosis of AA for specific subgroups of patients, including those with older age, diabetes mellitus and pediatric patients. Thirdly, most of the studies included in the present meta-analysis did not compare RIPASA and Alvarado scores with other diagnostic tests used in clinical practice. With these considerations in mind, the present systematic review and meta-analysis points out benefits and drawbacks of the two widely used scoring systems for the diagnosis of AA. Specifically, we found that the RIPASA scoring system can be useful both for excluding the diagnosis of AA and for relaying intermediate-risk patients to more accurate diagnostic imaging techniques. However, it is not currently possible to define a universal diagnostic test to be used in the clinical practice. The choice depends on several factors, including the resource to obtain data and different clinical settings. In this scenario, our findings could guide future studies to improve the current knowledge about the risk assessment of patients with AA, also promoting the implementation of existing scores and/or the development of innovative tools for clinical practice.

## Conclusions

In conclusion, the early diagnosis of patients with suspected AA is still a challenge for clinical practitioners and public health professionals. Although the existing scoring systems help in the risk assessment and in the prediction of clinical deterioration, these scores show variable values of specificity and sensitivity. In our study, the RIPASA score had a superior performance in identifying true positive patients, while the Alvarado score was better in predicting true negative patients. For this reason, further research should be encouraged to develop novel scores and strategies for improving the risk assessment of patients with suspected AA.

## Supporting information

**S1 Appendix. PRISMA 2020 checklist.**
(DOCX)

**S2 Appendix. QUADAS checklist.**
(DOCX)

## Author Contributions

**Conceptualization:** Guido Basile, Antonella Agodi.

**Data curation:** Giuliana Favara, Andrea Maugeri, Martina Barchitta, Andrea Ventura, Guido Basile.

**Writing – original draft:** Giuliana Favara, Andrea Maugeri, Martina Barchitta.

**Writing – review & editing:** Giuliana Favara, Andrea Maugeri, Martina Barchitta, Andrea Ventura, Guido Basile, Antonella Agodi.

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
