## [Decision Letter · Decision Letter 0]

4 Sep 2022

PONE-D-22-21559COMPARISON OF RIPASA AND ALVARADO SCORES FOR RISK ASSESSMENT OF ACUTE APPENDICITIS:  A SYSTEMATIC REVIEW AND META-ANALYSISPLOS ONE

Dear Dr. Agodi,

Thank you for submitting your manuscript to PLOS ONE. After careful consideration, we feel that it has merit but does not fully meet PLOS ONE’s publication criteria as it currently stands. Therefore, we invite you to submit a revised version of the manuscript that addresses the points raised during the review process.

ACADEMIC EDITOR:Is the protocol for the meta-analysis registered in any database like Prospero?Is there a particular reason for the authors not to assess for publication bias?

We look forward to receiving your revised manuscript.

Kind regards,

Ibrahim Umar Garzali, MBBS, FWACS

Academic Editor

PLOS ONE

Journal Requirements:

2. Please upload a new copy of Figures 1, 2, and 3 as the detail is not clear. Please follow the link for more information: https://blogs.plos.org/plos/2019/06/looking-good-tips-for-creating-your-plos-figures-graphics/

https://blogs.plos.org/plos/2019/06/looking-good-tips-for-creating-your-plos-figures-graphics/

Additional Editor Comments:

A very good meta-analysis. But there is need to mention if the protocol for the meta-analysis was registered in any database like prospero and if it was registered, provide the number.

Also there may be need to asses publication bias in some meta-analysis, why wasn't that done?

Reviewers' comments:

Reviewer's Responses to Questions

**Comments to the Author**

1. Is the manuscript technically sound, and do the data support the conclusions?

Reviewer #1: Yes

Reviewer #2: Yes

2. Has the statistical analysis been performed appropriately and rigorously? 

Reviewer #1: Yes

Reviewer #2: Yes

3. Have the authors made all data underlying the findings in their manuscript fully available?

Reviewer #1: Yes

Reviewer #2: Yes

4. Is the manuscript presented in an intelligible fashion and written in standard English?

Reviewer #1: Yes

Reviewer #2: Yes

5. Review Comments to the Author

Reviewer #1: The study topic is relevant to clinical practice, moderately well-written, clear and as a systematic review contributes to the body of literature on the topic of clinical diagnosis of acute appendicitis. It may go a long way in informing clinicians on what tool may or may not be superior in decision-making.

However, the following questions need to be clarified:

1. The authors mention 33 articles from 35 studies. While it may be normal to expect 35 articles from 33 studies as some studies can generate more than one articles, how were 35 studies reduced to 33 articles? Further, in the “Results” section while mentioning study design, the authors clarify 30 observational and 3 cross-sectional studies, making 33!

2. In discussing the individual studies included in the systematic review, the authors discuss NPV and PPV of RIPASA vis-à-vis Alvarado but only give the sensitivity and specificity values in the results of the meta analysis. Can the combined NPV and PPV be generated by the methods used to carry out the analysis because this may guide the clinicians further when comparing the two tools?

3. In the manuscript the authors admit that all the studies were done in a specific sub-set of global population. Furthermore, most of the studies that as argued by the authors used a higher cut-off for RIPASA/Alvarado to improve diagnostic parameters of the score are from a specific sub-set of the study groups. In authors opinion, what is the universal applicability of the results of this analysis and the studies reviewed, especially to the sub-set of global population not covered by the studies?

Reviewer #2: The study has highlighted the significance of RIPASA and Alvarado scoring systems and most importantly it has provided an insight into the diagnostic value of these test in terms of both their sensitivity and specificity. The authors however, the to address the following issues:

1.State the mean age ± SD

2. Discuss the implication of high sensitivity and low specificity of RIPASA as its relates to clinical practice; especially in diagnosis of fatal/severe conditions like AA and its role in reducing morbidity, mortality and healhtcare cost.

3. Kindly explain the statement that "In general the risk of bias was unclear or high in all domains.

6. PLOS authors have the option to publish the peer review history of their article (what does this mean?). If published, this will include your full peer review and any attached files.

Reviewer #1: No

Reviewer #2: No

---

## [Author Response · Author response to Decision Letter 0]

15 Sep 2022

Dear Editor, 

this document is intended for the convenience of the editor and reviewers and contains the list of the requested changes. We hereby submit to your attention a revised version of the manuscript in which we have considered all comments and suggestions. The following list of changes and answers to comments of Academic Editor and Reviewers addresses all changes made in the manuscript (in red font).

ACADEMIC EDITOR

AE: Is the protocol for the meta-analysis registered in any database like Prospero?

Authors: We would like to take this opportunity to thank the Academic Editor for his/her comments and suggestions which helped us in improving our manuscript. We apologize for not giving information on the registration in the PROSPERO database. Please consider details described in the revised version of our manuscript (lines 83-84).

AE: Is there a particular reason for the authors not to assess for publication bias?

A: We recognize that publication bias is one of the main source of misinterpretation of findings from meta-analyses. However, to our knowledge, there are no currently adequate methods to investigate publication bias in meta-analyses of diagnostic tests. For this reason, we have added this point as a main limitation of our work (lines 237-241).

REVIEWER 1 

R: The authors mention 33 articles from 35 studies. While it may be normal to expect 35 articles from 33 studies as some studies can generate more than one articles, how were 35 studies reduced to 33 articles? Further, in the “Results” section while mentioning study design, the authors clarify 30 observational and 3 cross-sectional studies, making 33!

A: We would like to take this opportunity to thank the Reviewer for his/her comments and suggestions which helped us in improving our manuscript.

We apologize if this point was not so clear. The meta-analysis included 33 studies. However, Abdelrhman et al. (2018) reported findings from two different populations, while Erdem et al. (2013) used two different couples of cut-offs for the RIPASA and ALVARADO scores. Accordingly, the meta-analysis was conducted on 35 different estimates of sensitivity and specificity. We have better described this point in lines 149-152.

R: In discussing the individual studies included in the systematic review, the authors discuss NPV and PPV of RIPASA vis-à-vis Alvarado but only give the sensitivity and specificity values in the results of the meta analysis. Can the combined NPV and PPV be generated by the methods used to carry out the analysis because this may guide the clinicians further when comparing the two tools?

A: We agree with this comment on the importance of NPV and PPV in interpreting the performance of a diagnostic test. Accordingly, we have reported the individual parameters for each study included in the meta-analysis. However, to our knowledge there are currently no methods to pool this kind of data through a meta-analysis. 

R: In the manuscript the authors admit that all the studies were done in a specific sub-set of global population. Furthermore, most of the studies that as argued by the authors used a higher cut-off for RIPASA/Alvarado to improve diagnostic parameters of the score are from a specific sub-set of the study groups. In authors opinion, what is the universal applicability of the results of this analysis and the studies reviewed, especially to the sub-set of global population not covered by the studies?

A: We are grateful for this comment that allowed us to discuss an important point of our study. Please consider changes in lines 251-255.

REVIEWER 2

R: The study has highlighted the significance of RIPASA and Alvarado scoring systems and most importantly it has provided an insight into the diagnostic value of these test in terms of both their sensitivity and specificity. The authors however, the to address the following issues: State the mean age ± SD

A: We would like to take this opportunity to thank again the Reviewer for his/her comments and suggestions which helped us in improving our manuscript. Since data on the mean age of individual studies are not reported uniformly, it is not possible to calculate the mean age and standard deviation of all included studies. 

R: Discuss the implication of high sensitivity and low specificity of RIPASA as its relates to clinical practice; especially in diagnosis of fatal/severe conditions like AA and its role in reducing morbidity, mortality and healhtcare cost.

A: We agree with this suggestion and thus we have discussed the implications in the discussion section. Please consider changes in lines 222-228.

R: Kindly explain the statement that "In general the risk of bias was unclear or high in all domains.

A: As requested, we have better explained this point in the limitation section (lines 234-237).

---

## [Editor Report · Decision Letter 1]

19 Sep 2022

COMPARISON OF RIPASA AND ALVARADO SCORES FOR RISK ASSESSMENT OF ACUTE APPENDICITIS:  A SYSTEMATIC REVIEW AND META-ANALYSIS

PONE-D-22-21559R1

Dear Dr. Agodi,

We’re pleased to inform you that your manuscript has been judged scientifically suitable for publication and will be formally accepted for publication once it meets all outstanding technical requirements.

Kind regards,

Ibrahim Umar Garzali, MBBS, FWACS

Academic Editor

PLOS ONE
---

## [Editor Report · Acceptance letter]

21 Sep 2022

PONE-D-22-21559R1 

COMPARISON OF RIPASA AND ALVARADO SCORES FOR RISK ASSESSMENT OF ACUTE APPENDICITIS:  A SYSTEMATIC REVIEW AND META-ANALYSIS 

Dear Dr. Agodi:

I'm pleased to inform you that your manuscript has been deemed suitable for publication in PLOS ONE. Congratulations! Your manuscript is now with our production department. 

Kind regards, 

on behalf of

Dr. Ibrahim Umar Garzali 

Academic Editor

PLOS ONE